# The Psychosocial Effect of Parental Cancer: Qualitative Interviews with Patients’ Dependent Children

**DOI:** 10.3390/children10010171

**Published:** 2023-01-15

**Authors:** Elise S. Alexander, Moira O’Connor, Georgia K. B. Halkett

**Affiliations:** 1Discipline of Psychology, School of Population Health, Faculty of Health Sciences, Curtin University, Bentley, WA 6102, Australia; 2Discipline of Psychology, School of Population Health/Curtin Health Innovation Research Institute (CHIRI), enABLE Institute, Faculty of Health Sciences, Curtin University, Bentley, WA 6102, Australia; 3Curtin School of Nursing/Curtin Health Innovation Research Institute (CHIRI), Faculty of Health Sciences, Curtin University, Bentley, WA 6102, Australia

**Keywords:** parental cancer, psychosocial, qualitative interviews, children, family

## Abstract

Background: Children living with parental cancer are at an increased risk for various psychosocial, emotional, and behavioural problems. However, research regarding how children are affected by their parent’s diagnosis is still developing and patients’ children are typically invisible in clinical practice. This study aimed to investigate how children are affected by their parent’s cancer diagnosis, from children’s perspectives. Methods: Informed by methods of grounded theory and embedded within a social constructivist framework, twelve children (ranging from 5 to 17 years) living with a parent with cancer were interviewed using a semi-structured format assisted by a novel approach derived from play- and art-based developmental literature. Results: Findings indicate that patients’ children are constantly worried and distressed, and there are barriers that can be overcome to mitigate this. Four overarching themes were identified: (I) Feeling worried and distressed; (II) Comprehending their parent’s cancer diagnosis; (III) Being disconnected from their supports; and (IV) Needing someone to talk to. Conclusions: Children experience considerable levels of ongoing worry and distress when a parent is diagnosed with cancer and have difficulties comprehending and articulating this. They also feel a level of disconnection from their usual support systems (e.g., parents) and are limited regarding who they can seek out and talk to. Mitigating children’s ongoing worries and distress by promoting the availability and accessibility of parents and other supports to children, and reducing communication barriers between children and adults, should be a primary focus of psycho-oncology research and practice.

## 1. Introduction

The five-year survival rates among Australian cancer patients aged 25 to 49 years are rising [1], meaning that patients and their families are living longer with the impact of a cancer diagnosis [2]. Global research indicates that many in this age range will be supporting dependent children while also coping with their diagnosis [3,4], presenting a major challenge for this cohort. In Australia, there are currently no population data regarding the prevalence and characteristics of children living with parental cancer. However, a longitudinal study conducted in Western Australia reported that between 1982 and 2015, 25,901 (approximately 24%) children (0–11 years) experienced a parent’s diagnosis of cancer [5].

When a parent is diagnosed with cancer, families are likely to experience disruptions to routines, relationship strains, changes in roles and responsibilities, financial pressures, and difficulty maintaining adequate social supports [6,7,8,9]. Families with dependent children face additional challenges, as parents report heightened levels of concern around how to appropriately support children, including apprehension about communication [10,11]. Children are also endeavouring to cope with, and adjust to, their parent’s cancer diagnosis and resultant family changes, while remaining on track developmentally [12,13,14,15]. The literature indicates factors such as age, gender, cancer stage, pre-existing comorbidities, parent’s marital status and psychological health are likely to influence how children are affected [16,17,18].

How children cope when a parent has cancer is an area of increasing research interest [13,19]. While some studies report resilience building and the potential for post traumatic growth among cancer patients’ children [15,20], there is evidence to suggest that the overall adjustment and emotional wellbeing of patients’ children is negatively affected and children are at risk of various maladaptive psychosocial, emotional, and behavioural stress responses such as somatic complaints, separation anxiety, levels of distress, confusion, rumination, worry, and intrusive thoughts [18,20,21]. For some children, these symptoms are likely to dissipate over time; however, there is evidence to indicate that other children remain vulnerable to ongoing long-term problems, including self-injury and post-traumatic stress symptoms [22,23]. This vulnerability may be associated with the level of disruption children experience in the initial stages following diagnosis [24].

Research indicates that cancer patients’ children prefer to be supported and informed by their parents [25]. Yet, a recent study found that parents are often overwhelmed by the cancer diagnosis [26] and find it challenging to foster conversations with children and distinguish what is typical developmental behaviour from indications that their child is struggling to adjust [27]. Parents tend also to wait for health professionals to broach the topic of children due to the tension between being in the patient role and the parental role, and the emotional challenges when discussing their children [10,28]. Health professionals are unlikely to raise questions about children with their patients [10,28] as they are typically inexperienced and have limited knowledge regarding children in families affected by cancer, and children are not visible or considered in adult clinical settings [29].

Alongside this, intervention research continues to focus on parent-proxy reports regarding the effectiveness of interventions among patients’ children [21,30], despite several studies highlighting that discrepancies are common between parents’ accounts and children’s [16,31,32]. Recent reviews further indicate that current programs and interventions are not effective in mitigating children’s various psychosocial and behavioural outcomes, including depression and anxiety [14,20,30]. However, qualitative findings regarding interventions and programs available for patients’ children support their feasibility and acceptability, indicating that there is a voracity for interventions among parents and children [14,30].

As discussed, the conflicting findings reported in the literature, (e.g., the different psychosocial outcomes reported among children) suggest that there are gaps regarding how children are affected by their parent’s cancer diagnosis and how they can be best supported (e.g., the diminished availability of parental support for their children). This study aimed to investigate children’s perspectives of how they are affected by their parents’ cancer diagnosis.

## 2. Methods

Ethics approval was received from Sir Charles Gairdner and Osborne Park Health Care Group HREC and Curtin University HREC, for the current study, in the approval of a broader study exploring how children are impacted by parental cancer, and involved interviews with health professionals [29], patients/parents [26], and children. This article focuses on children’s perspectives.

### 2.1. Design

This study was informed by principles of grounded theory and was positioned within a social constructivist framework [33]. This approach emphasises each individual’s constructed meaning of the world and unique experiences. This method was considered appropriate given the complexity of the research topic. Importantly, this methodology enables in-depth interpretation of how children construct meaning through their own social interactions and is flexible enough to be used among diverse populations, including young children of various ages, where adaptive methodologies are necessary [34,35]. Furthermore, it emphasises the co-construction of reality between the researcher and the participant [33]. This process is necessary with children, whose varying cognitive capacity makes it difficult for them to comprehend and articulate their experiences when a parent is diagnosed with cancer [29]. The steps taken in this study are depicted in Figure 1.

### 2.2. Participant Recruitment

Parents and patients (herein collectively referred to as parents) were recruited for interviews [26] and their children were also given the option to be interviewed. Recruitment took place at a tertiary teaching hospital in metropolitan Perth, WA using convenience and snowballing sampling. One researcher’s (MOC) pre-existing networks were used to approach hospital nurses who were asked to advise potential parents/patients of the prospective study. Fliers were also displayed on hospital bulletin boards. Inclusion criteria for parents/patients stipulated they must be a parent who had been diagnosed with cancer or a parent whose partner had been diagnosed with cancer, and that they had a child living with them aged 18 years or below. Interested parents contacted the primary researcher (EA) directly and a mutually convenient time and location for the interview was arranged. Children’s interviews were conducted independently of their parents. As new themes emerged, theoretical sampling was used to explore these further by approaching participants considered to have insight regarding novel themes. Recruitment, interviews, and analysis occurred concurrently (theoretical sampling) and continued until category saturation was achieved—that is, data were considered rich and detailed, and no new categories were emerging [36].

#### Participants

Twelve children were interviewed between April 2017 and June 2018. In total, 58% of participating children were female with a mean age of 9.2 (± 3.5) years (see Table 1). Most children were from an Australian background (75%) and spoke English as their first language. All children were attending school, ranging from pre-primary to year 11. 6 children from 3 different families, were siblings and participated in the study. There was an equal number of mothers and fathers who had been diagnosed with cancer. Cancer type varied, including bowel, brain, breast, lymphoma, melanoma, lung and oral.

### 2.3. Interviews

The researcher (EA, female) is a PhD (Psychology) candidate who has experienced the loss of a younger sibling to childhood leukemia in 2004, which prompted the researcher to pursue their studies in psychology, to address observed gaps in psychosocial support for children affected by cancer. Semi-structured interviews were used to enable the researcher to maintain topic consistency while also promoting alternative lines of enquiry, further explanation, and examples of topics where relevant. This also enabled the children to provide their own unrestricted perspectives. Interview questions focused on exploring how children perceived they were affected by their parent’s cancer diagnosis. The interview schedule (see Table 2) was guided by the research question, a general literature review, and findings from a systematic review [30]. Interviews spanned approximately 38 min (m = 37.13, SD = 21.77).

#### Children’s Activity

A novel approach was developed and used during the interviews to facilitate children’s capacity to articulate their responses to interview questions. This was influenced by arts, drawing, and projective techniques which are used in participatory research to assess children’s wellbeing [37] and distress among vulnerable child populations such as those affected by sexual abuse and war [38,39,40]. There is also evidence to suggest that these techniques are efficacious among children from different cultural backgrounds [41]. Informed by principles of play therapy, children were also provided with a series of age-appropriate toys (e.g., spinners and mini footballs) by the interviewer to facilitate discussion and rapport and provide a distraction from direct, face to face conversations [42]. They were then asked if they would like to draw a self-portrait alongside the researcher who also drew one. During this drawing exercise, the interviewer proceeded to ask children questions around their parent’s cancer diagnosis (see Table 1). Upon completion of the child’s self-portrait, they were asked to list any worries and concerns including, but not limited to, those related to their parent’s cancer diagnosis. As the child listed these, the interviewer wrote them down on coloured post-it notes. The child then positioned the post-it notes on their self-portrait. The more worry the child felt for each reported worry, the closer to the image of themselves it was placed. This can be observed in Figure 1, where one child demonstrated more worry about her mother’s cancer recurrence than about her mother missing out on activities.

### 2.4. Data Collection

Interviews were conducted in the participants’ homes (*n* = 7), at Curtin University (*n* = 2), temporary accommodation (*n* = 2), or the tertiary hospital (*n* = 1). Participants’ parents were provided with information sheets and written consent was gained. Additionally, children were provided developmentally appropriate information sheets and verbal assent was attained as per the National Statement on Ethical Conduct in Human Research chapter 4.2 [43]. Observational notes and journaling were used to record notable details regarding context and behaviours. All children completed their interview; however, 1 child refused to be recorded. A detailed summary was documented immediately following completion of their interview.

### 2.5. Data Analysis

Interviews were digitally recorded and transcribed verbatim. Transcribed data were analysed using methods of constant comparison [44] to identify themes. Several readings of transcripts by the primary researcher enabled data familiarity. Initial line by line coding of the first five transcripts focusing on gerunds (actions and processes) was used to develop codes [44]. These were then transferred into Microsoft Excel to index the data into manageable chunks and elevate these to form the basis of preliminary themes. Other members of the research team (MOC and GH) reviewed transcripts and themes were discussed and refined in an iterative process. Agreed upon themes were transferred to NVivo 12 where the remaining transcripts were coded, while remaining open to identifying further themes. During this process, the researcher remained open to identifying further themes. Memoing techniques were also used to support themes to be moved from the descriptive to the analytical level [33,45]. Guidelines and criteria outlined by Mays and Pope [46] and Braun and Clarke [47] were also followed to promote analysis rigor, including creating an audit trail of the methods and data analysis used and providing transparent and accurate reports of the research studies and findings. The consolidated criteria for reporting qualitative research (COREQ) guidelines [48] were used to promote study quality and reporting rigor.

## 3. Findings

Findings revealed that when a parent was diagnosed with cancer, their children experienced heightened levels of worry and distress, that was often persistent and intrusive. Children felt they needed someone to talk to, to help them comprehend their experiences and articulate themselves to others, including adults. However, there were barriers to communication during this critical time, and children often felt disconnected from usual support systems, including their parents and friends. Four overarching themes were identified: (I) Feeling worried and distressed; (II) Comprehending their parent’s cancer diagnosis; (III) Being disconnected from their supports; and (IV) Needing someone to talk to. Talking about their parent’s cancer diagnosis, was a subtheme identified under comprehending their parent’s cancer diagnosis. These themes are explained in more detail.

### 3.1. Feeling Worried and Distressed

During a parental diagnosis of cancer, children experienced ongoing distress and worry related to their parent’s cancer diagnosis, alongside worries commonly experienced by children whose parent is not ill (e.g., peer relationships and academic performance). For example, patients’ children were likely to harbour worry and concern regarding how their parent was coping, and possible outcomes including recurrence or death. This is illustrated in Figure 2, a drawing by a child who identified her mother’s cancer recurrence and death, alongside missing out on school, as her three most prevalent worries.

Children’s worries tended to be ongoing, and their level of intrusiveness appeared vulnerable to external circumstances such as whether the parent was in hospital or actively receiving treatment. For example, one child discussed her experience of an activity facilitated by an arts therapist who had been engaged by the family. In this activity, she was asked to report her worries and feelings. Her reiteration of this highlighted the chronic nature of her worry and the direct impact her mother’s fluctuating health status had on her levels of happiness.


*Participant: “In my ‘worries’ I usually write about mummy’s cancer and in the ‘feelings’ I usually write worried, happy, angry and frustrated”.*



*Interviewer: “Is that how you generally feel?”*



*Participant: “Yep”.*



*Interviewer: “You feel worried and frustrated a lot of time?”*



*Participant: “Yes”.*



*Interviewer: “When do you feel happy?”*



*Participant: “When mummy’s okay and she’s doing stuff”*
(Batari; female: 8.5 years)

Often, this worry also generalised to other family members (including the healthy parent), friends, and pets.


*“I worry about the dogs dying. I worry about mum dying. I worry about all of my family, really”*
(Kayla, female: 10 years).

This participant also evidenced this as her most prevalent worry in her drawings, as observed in Figure 3.

As evidenced in Figure 4, some children also worried about the likelihood of developing cancer. In this drawing, the proximity of the worry ‘getting cancer’ to the portrait, indicated that their own cancer risk was their most prevalent concern.

Often, children’s worries led to increased vigilance regarding their parent’s health. This tended to impact negatively on them, particularly those whose parent was an outpatient, lending children to be increasingly exposed to the disease and treatment side-effects.


*“I just want to go home every day... Because I want to stay with my mummy to make her feel better”.*
(Arianna; female: 6.5 years).

Moreover, this worry persisted even when the patient was well or in remission.


*Interviewer: “How did you deal with that worry?”*



*Participant: “I don’t think I really dealt with that worry. It sought of just lingered around”.*
(Lucas; male: 12 years).

When this same child (Lucas) was asked about how this worry affected him, he stated, “I don’t think it had much of an effect on me then. I feel like it has more of an effect on me now because I know that my mum survived and what could have happened and what she did to keep it all [life] going”.

### 3.2. Comprehending Their Parent’s Cancer Diagnosis

Children’s awareness of their parent’s cancer diagnosis and their capacity to comprehend this and other related information, varied. However, most children indicated that they knew very little about this.


*“I don’t know about it [cancer]... I just know that cancer is a bit dangerous… Because people that have cancer may die...I don’t really know much about what sort of cancer she had or how she got saved. I just know that she had cancer and she was lucky enough to get saved”.*
(Indigo; female: 8 years).

These children often referred to earlier experiences or knowledge to ‘fill in the gaps’, particularly if someone they knew had previously had cancer. However, this sometimes led to the formation of misconceptions about the disease.


*Interviewer: “Can you tell me what you know about mum’s cancer?”*



*Participant: “Brain cancer, kills people”*
(notably, the parent did not have a brain cancer diagnosis).


*Interviewer: “Did someone you know have brain cancer?”.*



*Participant: “It’s granddad. He died”.*
(Arianna; female: 6.5 years).

However, some children indicated that they had no questions regarding their parent’s diagnosis, and one child stated he preferred not to be further informed.


*“No. I know that it’s not life-threatening and that it’s dying slowly. It’s minimizing. So, that’s all I want to know”.*
(Darius; male: 13 years).

#### Talking about Their Parent’s Cancer Diagnosis

Children’s difficulties in comprehending their parent’s cancer diagnosis appeared to also affect their capacity to talk about their parent’s cancer diagnosis and articulate any thoughts, emotions, and questions they had about this. Many children (including older children) struggled to answer questions in more depth than ‘yes or no’ responses even when probed as per the interview schedule. Hence, they required ongoing discussion and support to drill down further and unpack what they knew about their parent’s diagnosis.


*Interviewer: “Do you know anything else about mum’s cancer?”.*



*Participant: “No”.*



*Interviewer: “Do you know if she’s getting any medication for it?”.*



*Participant: “No”.*



*Interviewer: “Are they giving her anything to make her feel better?”.*



*Participant: “Yeah. Medicine”.*
(Arianna, female: 6.5 years).

One child indicated that he regretted not asking more questions at the time; however, he conceded that he had limited awareness of the diagnosis at the time and was unsure of what he needed to know, what he needed to ask, and how to ask.


*Participant: “When I look back it, I wished I’d asked more questions… I’d feel a sense of closure if I did ask…”.*



*Interviewer: “Do you know what you would ask?”.*



*Participant: “I’m generally unsure of it, I just feel the need to know something”.*



*Interviewer: “Is it something you can ask mum about?”.*



*Participant: “It might be, but I’m unsure of how to do this”.*
(Lucas, male: 12 years).

All children required some level of facilitation by the interviewer to promote discussion. One child was particularly reluctant to speak. However, upon engaging in play and then moving to the children’s activity (detailed in the methods section), he was encouraged to convey his understanding about his mother’s diagnosis. This is evidenced in Figure 5, where he indicated the primary cancer site of his mother’s cancer diagnosis.

### 3.3. Being Disconnected from Their Supports

Most children reported a loss of quality time spent with their ill parent.


*Participant: “I spent more time with her before she got sick. She’s having another operation to take the bag away and then we’re going to have more time to be with her again”.*



*Interviewer: “Are you looking forward to that?”.*



*Participant: “I’ve been waiting for it for 1000 years”*
(Arianna, female: 6.5 years).

They also reported their parents’ difficulties to take care of their basic needs such as preparing meals.


*“When she got cancer, she couldn’t do it [cooking] so she just gives us like baked beans or spaghetti”.*
(Darius, male: 13 years).

Children also reported they had less opportunity to spend time with other support networks including extended family.


*“Mummy can’t drive that much so we can’t really go down to (Location A) and (Location B) that much. So, I don’t get to see my family because most of them live in (Location A)”.*
(Batari, female: 8.5 years).

Many were unable to see friends during or outside of school.


*“I miss my friends because most of them, they are not at home and we’re not going to school”.*
(Arianna, female: 6.5 years).

Most children were also required to give up sports and hobbies due to their parent’s cancer diagnosis.


*“Well, now I can’t exactly go out on walks with the dog because Dad used to come with me and now, he can’t drive me to dancing. I used to do dancing, tap, acro and I started jazz”.*
(Sarah, female: 11 years).

Some children experienced greater upheaval and disruption to their lives, including integral relationships and support networks. For example, one family migrated to Australia for treatment, leaving their father, extended family, friends, school, and all other support networks.


*Participant: “The one that made me happy really, really, really happy was my grandpa and my favourite auntie”.*



*Interviewer: “In [home country]?”*



*Participant: “Yep”.*



*Interviewer: “So, you spent a lot of time with them, did you?”*



*Participant: “Yep”.*
(Indigo; female: 8 years).

Another family was required to relocate immediately following diagnosis from regional to metro WA to receive treatment. Hence, there was a significant shift in the children’s support networks available to them, with family and friends no longer close.


*Interviewer: “When you found out that daddy was sick, and your family moved away, how did you feel then?”*



*Participant: “Worried... “That’s when Mr. Worry Monster came. Until I made some friends”.*
(Farrah; female: 7 years).

### 3.4. Needing Someone to Talk to

Children preferred to ask their parents any questions about the cancer diagnosis and, if parents were not available, then they would consider a teacher or other adults they perceived as knowledgeable (e.g., grandparent).


*“I would ask my Mum, or I could ask my dad. I would ask either one of them or maybe someone who was in the house. If I was with my grandma and granddad, I would ask them as well. I’m comfortable with asking anyone that is older than me, not my friends because they probably wouldn’t know as much as I would”*
(Sarah; female: 11 years).

However, it appeared children did not have many options for talking to someone other than their parents. One child said he would be unlikely to chat to or ask questions of teachers or health professionals, indicating a perceived ‘gap’ between himself and these supports.


*Participant: “They all repeated the same thing, ‘if there’s anything you want to tell us, you can tell us…’. I genuinely don’t think that works…this doesn’t always fill the gap”.*



*Interviewer: “What’s the gap?”.*



*Participant: “The gap is a feeling of emptiness, teachers saying you can get something off your chest is a feeling that it’s not enough, there’s a void between you and them that doesn’t make it feel like you can talk to them”*
(Lucas; male: 12 years).

As observed in Figure 6, this was further supported by the child’s drawing which indicated that this gap identified between the child and health professionals was something that concerned him.

One child reported that she would ask her play therapist any questions; however, when prompted, she indicated that she had little opportunity to ask questions during the sessions.


*“The therapist only goes for 10 min—she asks questions. I don’t really get a chance to ask”.*
(Batari; female: 8.5 years).

When asked if they would talk to their friends, most children indicated that their friends were unlikely to understand or empathise with their situation, therefore they preferred not to.


*“I would talk to my friends, but I can’t ask them questions because I don’t think they would understand them”.*
(Sarah; female: 11 years).

Many children indicated that they wanted to talk to someone about what they were going through. As observed in Figure 5, one child reported the need for an individual to help him process and ‘express’ complex issues such as managing extreme physiological sensations associated with anxiety and worry (e.g., racing heart) and cognitions around what would happen to his mother including radical changes due to the cancer diagnosis and treatment. Some children were also welcoming of meeting other children whose parent also had cancer.


*“Sometimes yes, because they would be more likely to understand than some of my friends who have no family problems with that. Although they do try to help me, my actual friends, and they try to understand as much as possible, but sometimes you just can’t understand”*
(Sarah; female: 11 years).

## 4. Discussion

This qualitative study investigated children’s perspectives of their parent’s cancer diagnosis and how it affected them. Children discussed feeling worried and distressed; trying to understand, comprehend and talk about their parent’s cancer diagnosis; being disconnected; and needing someone to talk to.

In this study, children reported heightened levels of ongoing worry and distress associated with their parent’s cancer diagnosis. They worried about parents’ symptoms and disease outcome, treatment, recurrence, and death. While it is common for all children to experience thoughts around death and illness and be exposed to such concepts [49], these thoughts were experienced as very intrusive for cancer patients’ children. Patients’ children bear witness to the physical and cognitive changes associated with the disease and treatment, which exacerbates their worries and distress; and there is a real possibility death may occur [50]. There was tendency for children’s worries to generalise to family and friends, and for children to worry about their own likelihood of being diagnosed with cancer, which in some instances may be warranted based on the child’s previous experiences with cancer and the possibility of hereditary [51]. This appeared to be exacerbated for children whose parent was an outpatient and they were increasingly exposed to physical changes and had trouble escaping the impact. There is evidence for long-term consequences associated with even low levels of chronic stress in developing children, such as symptoms of Post-Traumatic Stress Disorder, anxiety and depression, obesity, and alcohol and substance abuse (see review by Wiss and Brewerton [52]), which further supports suggestions that patients’ children could remain vulnerable later in life [18,22,23]. Our findings support previous reports elucidating children’s ongoing worry, distress, and threat to health-related quality of life, as imperative adverse challenges experienced by patients’ children, requiring further research and clinical support [53].

Children in this study demonstrated varying levels of knowledge and awareness regarding their parent’s diagnosis; however, this was generally limited. Most children demonstrated difficulties comprehending their parent’s diagnosis and disease related information associated with this and struggled with articulating and unpacking their thoughts and emotions around this. In the absence of information, children were likely to construct their own meanings and answers, which often lead to misconstrued or inaccurate cognitions, and even magical thinking. Our findings are consistent with those reported in previous studies that have explored the impact of a parent’s cancer diagnosis on children [54,55,56]. Open, timely, and age-appropriate communication with patients’ children is imperative in the parental cancer literature, to mitigate children’s levels of distress and resolve cognitive inaccuracies or perceptions of insufficient information provision, and to support children with adapting during their parent’s diagnosis [14,18,21,50]. Yet, adults tend to focus on the more complex biological details of the diagnosis when speaking to children, and children find it difficult communicating their emotional problems to parents [57]. Parents and children’s appeal for guidance regarding communication continues to be well documented in the literature, (see review by Walczak et al. [20]), however, remains an unresolved area of need among families.

While children in this study preferred to talk to and be supported by their parents, parents were often unavailable and they were limited regarding who they could seek out, including extended family and friends who were difficult to access or considered unlikely to understand. This is commonly reported in the literature (see review by Morris et al., [18]); however, concerningly parents are often overwhelmed by their own experiences and challenges associated with the cancer diagnosis, making their physical and mental availability to children challenging [10,26]. Furthermore, research evidences that the parent–child relationship and family dynamics are integral to maintaining children’s wellbeing, with parental psychological health being inextricably associated with children’s [58,59]. While some children in this study considered speaking to other children in similar circumstances, interventions of this nature (e.g., Children’s Lives Include Moments of Bravery (CLIMB), [60]; The On Belay Program, [61]) are yet to prove effective at mitigating children’s emotional and behaviour problems, particularly in the long term [14,30]. Research that connects children with extended supports remains imperative and should continue to be an integral part of intervention research; however, future research should also continue to prioritise supporting the physical and mental capacity and accessibility of parents, to promote their availability to children [26].

Alternatively, health professionals who formed part of the parent’s oncological team were either not considered or too were perceived as inaccessible by most children who were reluctant to approach them. Yet, there is suggestion in the literature that health professionals are well placed to identify or support cancer patients’ children and provide a gateway entry into clinical systems [10,29]. Health professionals are also reluctant to approach patients’ children for reasons including a lack of knowledge or confidence in doing so, as the primary focus of their care is the patient, and their expertise is adult based [29,62]. Moreover, children are generally protected and kept from clinical settings by the patient [10,26], hence exacerbating patients’ children’s invisibility to health professionals and clinical systems. This appears to be a systemic gap whereby intervention, such as a family support worker or furthering health professionals’ education, could promote health professionals and children’s capacity to seek out one another, thus alleviating this burden on overwhelmed parents.

In this study, all children required some level of support or resources to assist them with talking about their parent’s cancer diagnosis. Expressing their thoughts and feelings appeared to be challenging, and most children were avoidant or deflective of this. Even older children reported their difficulties with expressing their worries. The inclusion of play and arts-based activities helped children in this study construct and express their thoughts and emotions. For some reluctant children, these approaches facilitated discussion; for others, it helped them comprehend and articulate their thoughts. This approach also enabled the researcher to drill down further regarding children’s meaning behind their verbal responses. This aligns with current literature evidencing fundamental differences in communication techniques, and comprehension levels, between children and adults, yet there is a continued expectation for children to communicate at an adult level [63,64]. The literature on communicating with children advocates for such innovative approaches as those used in this study [63,64]. Increasing recognition for including children in research also highlights the need to develop tools, resources, and guidelines that promote effective communication with children [65].

### 4.1. Limitations

It is important that these reported findings are interpreted with limitations of this study in mind. The age range of participants was broad; focusing on a narrower age range may have yielded different findings, and this could be considered in future studies. Participants were recruited through opt in methods and predominantly through cancer support services and Comprehensive Cancer Centres. Therefore, it is likely the participants in this sample came from families/parents with a greater psychological awareness and health literacy. This is commonly reported in similar studies [66,67,68,69]; however, a sample from a more diverse background, respectively, may be warranted. The interview techniques used were novel and developmentally appropriate; replication of this approach appears warranted to enhance authenticity.

### 4.2. Clinical Significance

The findings of this study can be used to inform current approaches to communicating and consulting with children living with parental cancer. Children are also disconnected from other supports outside of their parents, which is concerning as parents are often not positioned to attend to their children’s needs due to the challenges associated with the disease and treatment. Hence, our findings can also be used to inform guidelines and practice in psycho-oncology and intervention research by promoting the need for parents to be adequately supported, and for children to be appropriately connected with supports and resources outside of their family where necessary.

## 5. Conclusions

Children living with parental cancer are at heightened risk for psychosocial, emotional, and behavioural problems. Despite increasing research interest regarding how these children are affected, there is little consultation with children themselves. This is concerning considering the documented inaccuracies between child self and parent proxy-reports. Interviews with patients’ children revealed that children experience heightened levels of ongoing worry and distress when a parent is diagnosed with cancer and feel a level of disconnection from usual support systems, including parents and friends, during this critical time. Comprehension and communication barriers identified in this study can be used to inform future intervention research and clinical practice to enable children and adults to effectively communicate with each other.

Highlights from this study:Cancer patients’ children experience heightened levels of worry and distress at the time of diagnosis, and this continues to affect some children’s psychosocial wellbeing past their parents’ remission and/or bereavement.Children feel disconnected from their available support networks, including parents who are often unavailable or pre-occupied, extended family and friends who are difficult to access and perceived unlikely to understand, and health professionals who they do not consider for psychosocial and emotional support.Children need help with comprehending their parents’ cancer diagnosis and their associated complex thoughts and emotions, which impacts their capacity to effectively communicate with parents and other adults.

## Figures and Tables

**Figure 1 children-10-00171-f001:**
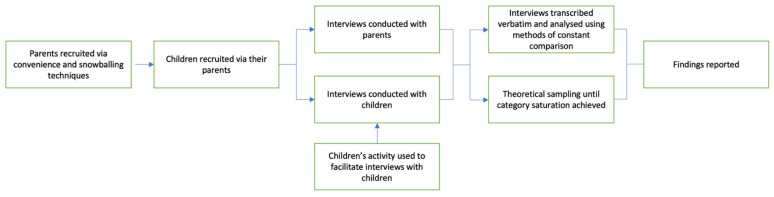
Methods undertaken in the present study.

**Figure 2 children-10-00171-f002:**
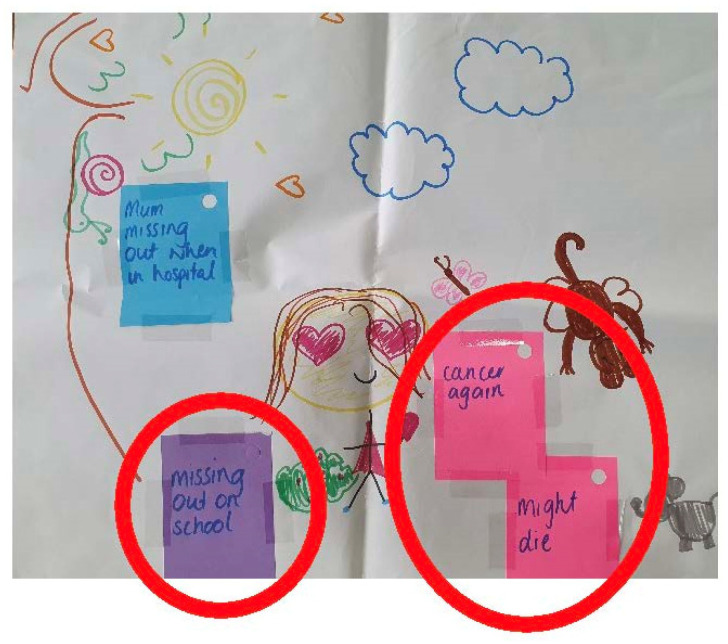
Child’s self-portrait and relative worries (Indigo; female: 8 years).

**Figure 3 children-10-00171-f003:**
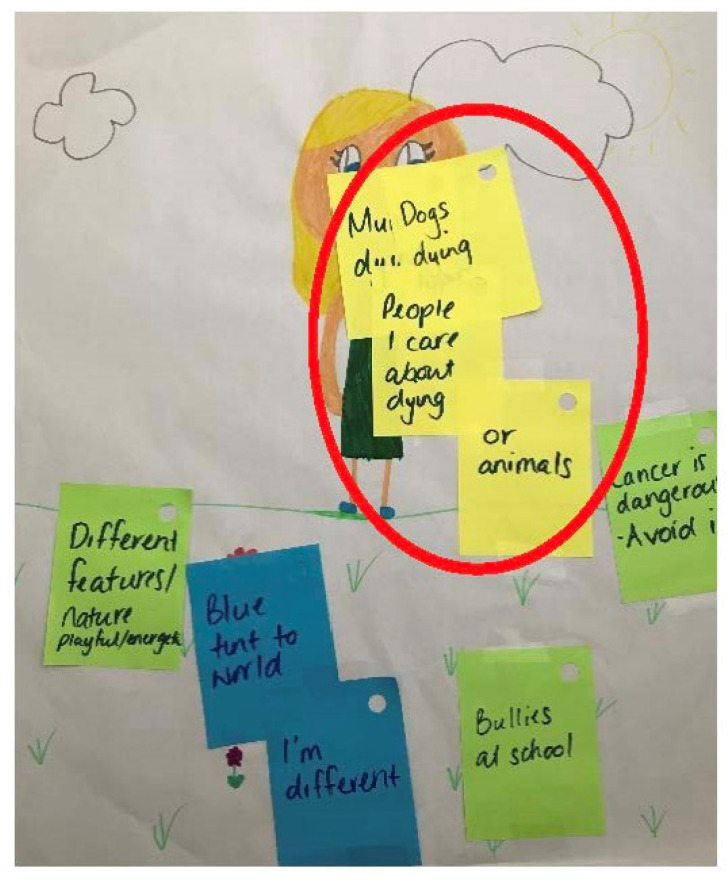
Child’s self-portrait and relative worries (Kayla; female: 10 years).

**Figure 4 children-10-00171-f004:**
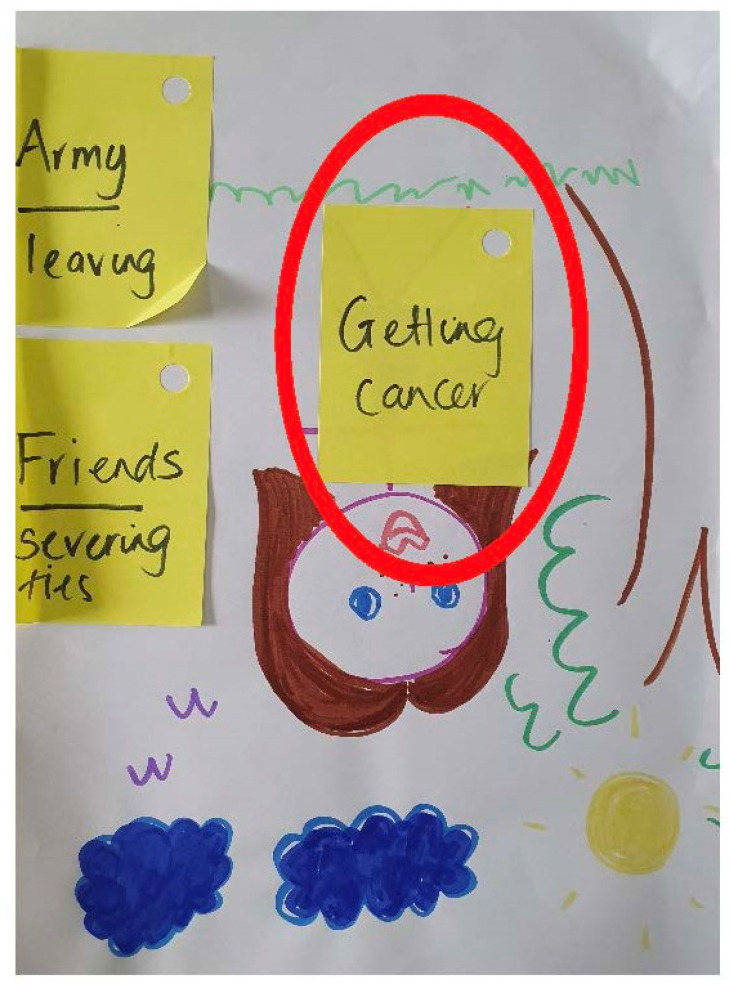
Child’s relative worries (Huxley; male*: 17 years). * not a self-portrait, this participant preferred to use the researcher’s drawing.

**Figure 5 children-10-00171-f005:**
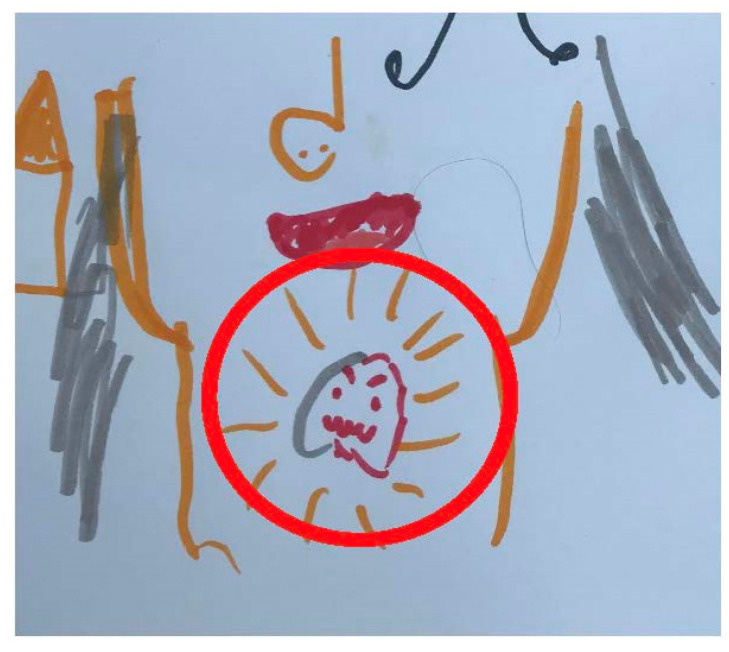
Child’s drawing of parent’s cancer site (James; male: 6 years).

**Figure 6 children-10-00171-f006:**
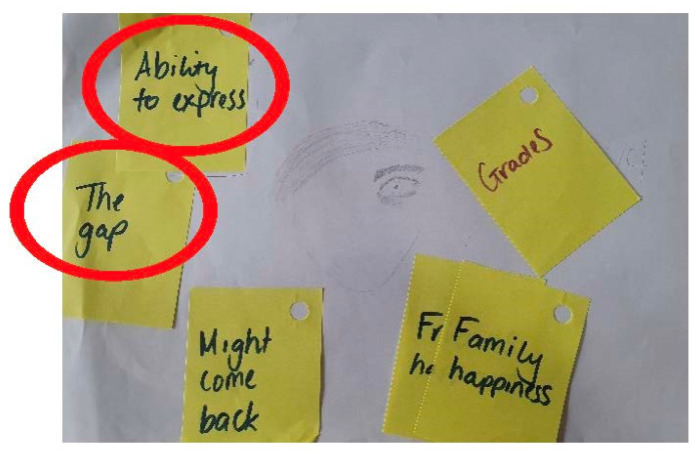
Child’s self-portrait and relative worries (Lucas; male: 12 years).

**Table 1 children-10-00171-t001:** Demographics.

Children	Number of Participants	*n* = 12
Age	Range Mean age (SD)	5–17 years 9.46 (±3.43) years
Gender	Female Male	58% or *n* = 7 42% or *n* = 5
Cultural background	Australian	75% or *n* = 9
	Indonesian	17% or *n* = 2
	Malaysian	8% or *n* = 1
Parent with cancer	Mother	50% or *n* = 6 *
	Father	50% or *n* = 6*
Parent’s primary cancer diagnosis **	Bowel cancer	2
	Brain	1
	Breast	1
	Burkitt’s lymphoma	1
	Lymphoma	1
	Melanoma	1
	Non-Hodgkin’s Lymphoma B cell	1
	Lung	1
	Oral	1
Stage **(at time of interview)	II	3
	III	1
	IV	3
	Not reported/remission/deceased	3

* *n* = 3 sets of siblings (*n* = 7 participants); siblings’ parent with cancer was counted multiple times. ** total number of patients included in the study was *n* = 10 (for further details regarding parents and patients, see Alexander et al., in review).

**Table 2 children-10-00171-t002:** Topic guide for qualitative interviews.

Number	Question	Prompts
1	Can you tell me about your family?	Such as who is in your family? Do you have any pets?
2	What are the fun things your family enjoy doing together?	Have any of these things changed lately?
3	Is there anything that you worry about?	
4	I was hoping you could tell me a little bit about your [mum/dad]. Has [mum/dad] been sick lately?	
5	What do you call [mum’s/dad’s] sick/sickness?	
6	Tell me what you know about [mum’s/dad’s] sickness?	
7	If you have a question about [mum’s/dad’s] sickness, who do you ask or what do you do?	
8	Is mum and dad OK talking to you about [mum/dad] not being well?	[If yes] Tell me some of the things you talk about with mum and dad? [If no] Would you like to be able to talk to mum and dad about this more?
9	Are there more things you want to know about [mum’s/dad’s] sickness?	[If yes] Tell me what sort of things?
	What are some things you do to help you feel better about [mum/dad] not being well?	
	Has life been different since [mum/dad] found out [he/she] was not well?	[If yes] Tell me how it has been different?
	Are things still the same with your friends, or have they changed?	[If they have changed] Tell me how they have changed?
	Are things still the same at school, or have they changed?	[If they have changed] Tell me how they have changed?
	Is there someone at school you prefer to talk to about [mum/dad] not being well?	[If yes] Tell me who this person is?
	What makes you feel the happiest lately?	[Prompt] Activities? Things? Items? People?
	And, what makes you feel unhappy or sad lately?	[Prompt] Activities? Things? Items? People?
	If I asked you to do a special activity with mum or dad, and it could be any kind of activity, what would that special activity be?	
	If you had a friend that found out their mum or dad was not well in a similar way to your [mum/dad], what would you do to help that friend?	
	If you had 3 wishes, what would those wishes be?	

## Data Availability

Data are available upon request.

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
