# Peer review of "The Psychosocial Effect of Parental Cancer: Qualitative Interviews with Patients’ Dependent Children"

_children, 2023, doi:10.3390/children10010171_

Round 1

Reviewer 1 Report

This is an important topic of research.  Please see suggested revision below.

Introduction is well-developed and presents current evidence related to the topic.  In the last paragraph, the authors state that gaps remain and then proceed with the aim. I suggest adding a few sentences here to identify where are these gaps?

Methods

Th authors describe the design as being guided by grounded theory.  however, there is no evidence of grounded theory in this manuscript.  The purpose of GT is to develop a substantive theory.  The hallmark of GT is a substantive theory describing the underlying social processes grounded in the data.  This process was not described nor was there a substantive theory developed.  The methods and approach described seems to be based on a thematic analysis. If a constructivist framework was used, greater detail is needed to support this claim. Specifically, the authors need to clarify the process used and design used to guide this research.  

Data analysis: "Transcribed data were analysed using methods of constant comparison [42] to identify themes".  then the authors proceed to identify Braun and Clarke as a guide to increase rigor.  Braun and Clarke do not use constant comparison and have 6 specific steps for data analysis. As such, the data analysis process seems contradicting.  Please add more details as to how the analysis was conducted in each of Braun and Clarke's steps.  How was the data from the art analyzed?  

The discussion of Rigor is completely missing.  Therefore, it is difficult to assess the quality of the study. 

Overall, there seems to be a disconnect between the method of analysis outlined by a GT process and the methods presented in the manuscript.

Discussion:  Important literature was identified in relations to the findings but the authors state that the themes  " worry", talking about diagnosis, being disconnected and needed support from parents" were all findings confirmed by other research studies.  However, the data presented was unique but the authors did not present or talk about the uniqueness of the data.  Specifically, there is no mention of the art which brings a unique voice to the data and discussion.  Suggest revising discussion to include how these data bring forward new evidence.

Practice Implications:  A few implications were mentioned.  However, the authors state the children should be connected to outside resources.  Although this may be an option, the best support for these children is their parents.  I suggest adding implications that include ideas of how to support parents to communicate and connect with their children while going through treatment. 

Reviewer 2 Report

This paper is very interesting and has great quality. It is a great contribution to scientific knowledge in its field. It uses an approach that is very important as it complements biomedical and clinical information. In addition, the paper is written in a very clear way and helps the reader a lot to follow the steps of the research. Nevertheless, I believe that there are small aspects that could be improved.

In the methodological section it would be convenient to introduce a flow chart to help the reader to understand the procedure in a more visual way.

I think it would be convenient to introduce, at the beginning of the results, a brief paragraph summarizing the main categories found. It would also be convenient to summarize, if possible, the codes detected in the work. In addition, I suggest introducing a graph to clarify this.

At the bottom of page 13 and on page 14 the transcriptions should be in italics.

The conclusions should be further expanded by indicating the main highlights of paper.
